# Causes of Vitamin K Deficiency in Patients on Haemodialysis

**DOI:** 10.3390/nu12092513

**Published:** 2020-08-20

**Authors:** Signe Wikstrøm, Katrine Aagaard Lentz, Ditte Hansen, Lars Melholt Rasmussen, Jette Jakobsen, Henrik Post Hansen, Jens Rikardt Andersen

**Affiliations:** 1Department of Nutrition, Exercise and Sports, University of Copenhagen, Rolighedsvej 26, DK-1958 FC Copenhagen, Denmark; siw93@outlook.com (S.W.); katrineaalentz@gmail.com (K.A.L.); 2Department of Nephrology, Herlev-Gentofte Hospital, University of Copenhagen, Borgmester Ib Juuls Vej 1, DK-2730 Herlev, Denmark; Henrik.Post.Hansen@regionh.dk; 3Department of Clinical Medicine, University of Copenhagen, Blegdamsvej 3B, DK-2200 Copenhagen, Denmark; 4Department of Clinical Biochemistry and Pharmacology, Odense University Hospital, DK-5000 Odense, Denmark; Lars.Melholt.Rasmussen@rsyd.dk; 5National Food Institute, Technical University of Denmark, DK-2800 Lyngby, Denmark; jeja@food.dtu.dk

**Keywords:** haemodialysis, vitamin K, phylloquinone, menaquinone, dp-ucMGP, D-xylose test

## Abstract

*Background:* A low vitamin K status is common in patients on haemodialysis, and this is considered one of the reasons for the accelerated atherosclerosis in these patients. The vitamin is essential in activation of the protein Matrix Gla Protein (MGP), and the inactive form, dp-ucMGP, is used to measure vitamin K status. The purpose of this study was to investigate possible underlying causes of low vitamin K status, which could potentially be low intake, washout during dialysis or inhibited absorption capacity. Moreover, the aim was to investigate whether the biomarker dp-ucMGP is affected in these patients. *Method:* Vitamin K intake was assessed by a Food Frequency Questionnaire (FFQ) and absorption capacity by means of D-xylose testing. dp-ucMGP was measured in plasma before and after dialysis, and phylloquinine (vitamin K_1_) and dp-ucMGP were measured in the dialysate. Changes in dp-ucMGP were measured after 14 days of protein supplementation. *Results:* All patients had plasma dp-ucMGP above 750 pmol/L, and a low intake of vitamin K. The absorption capacity was normal. The difference in dp-ucMGP before and after dialysis was −1022 pmol/L (*p* < 0.001). Vitamin K_1_ was not present in the dialysate but dp-ucMGP was at a high concentration. The change in dp-ucMGP before and after protein supplementation was −165 pmol/L (*p* = 0.06). *Conclusion:* All patients had vitamin K deficiency. The reason for the low vitamin K status is not due to removal of vitamin K during dialysis or decreased absorption but is plausibly due to a low intake of vitamin K in food. dp-ucMGP is washed out during dialysis, but not affected by protein intake to a clinically relevant degree.

## 1. Introduction

Vitamin K is a fat-soluble vitamin primarily present in green vegetables and dairy products. Vitamin K exists in three forms—phylloquinone (K1), menaquinones (K2) and menadione (K3) [1]. Phylloquinone is mainly present in vegetables and herbs and is the main source of vitamin K in the diet. Menaquinones are synthesized by bacteria and are present in fermented foods and are also synthesized in the large intestine as well. Unlike phylloquinone, menaquinones are further divided into subgroups depending on the number of isoprenoid groups in the sidechain. The third form of menadione (vitamin K_3_) is a synthetic analogue primary used in feed, but also an intermediate in metabolism of phylloquinone to menaquinone-4 (MK4) [2].

Vitamin K is essential for activation of proteins containing the amino acid glutamine, including Matrix Gla Protein (MGP). MGP is a protein with 84 amino acids and 10–14 kDa [3,4,5]. Vitamin K activates MGP by carboxylating the glu sidechains of MGP to Gla. Because of vitamin K’s function in MGP activation, dephosphorylated-uncarboxylated MGP (dp-ucMGP) is a measure of vitamin K status, as an increase in dp-ucMGP reflects vitamin K deficiency [6]. Active MGP, but not inactive dp-ucMGP, has been suggested to be an inhibitor of vascular calcification [7,8,9,10].

Patients with chronic kidney disease (CKD) on haemodialysis have a low vitamin K status [11,12,13] which responds to vitamin K supplementation [14,15,16]. A low intake of vitamin K is associated with vascular calcifications, and a high risk of cardiovascular disease [11,17].

The underlying cause of vitamin K deficiency in patients on haemodialysis is not known. Previous studies indicate that a low intake of foods rich in vitamin K may be an underlying cause, but indications are rather speculative as restrictions of phosphate- and potassium-containing foods automatically lead to a low intake of vitamin K [3,14].

No published studies have investigated other underlying causes of vitamin K deficiency than the intake of vitamin K in haemodialysis patients. Therefore, the aim of this study was to look for alternative explanations or contributors to the problem. This included washout of vitamin K (phylloquinone) during haemodialysis and reduced absorption capacity. Factors influencing the regulation of the biomarker dp-ucMGP were also examined, as well as effects of protein intake and washout of dp-ucMGP during dialysis.

## 2. Materials and Methods

This study consisted of four substudies:Vitamin K status and intake,Washout of vitamin K (phylloquinone) and dp-ucMGP during haemodialysis,Absorption capacity of vitamin K in the small intestine, andThe effect of protein intake on dp-ucMGP.

All patients treated in the haemodialysis unit at Herlev Hospital, Herlev, Denmark, were screened according to the in- and exclusion criteria. Patients older than 18 years and who had been on haemodialysis for more than three months were included. Exclusion criteria were warfarin treatment, known malabsorptive diseases and recent intake of vitamin K supplements. For substudy 3, diabetes mellitus was an exclusion criterion as well. Some participants were included in more than one substudy, but every substudy was carried out separately.

### 2.1. Vitamin K Status and Intake

Thirty patients answered a Food Frequency Questionnaire (FFQ) about foods containing phylloquinone and/or menaquinones. The questionnaires were completed during the haemodialysis session. Blood was drawn immediately before dialysis start and EDTA plasma was used to measure dp-ucMGP.

The Vitamin K FFQ was modified to Danish based on information on phylloquinone and MK-4 in the USDA Food Composition Databases [18] and the data from a Dutch study investigating the contents of phylloquinone and MK-4 to MK-9 in foods [19]. Foods were excluded from the calculations if they were ingested infrequently or had a vitamin K content below 20 μg/100 g.

### 2.2. Washout of Vitamin K_1_ and dp-ucMGP

Sixteen participants had blood drawn before and after haemodialysis to measure dp-ucMGP changes in plasma during a dialysis session. Meanwhile, the individual dialysate was collected 20–30 min after start and 20–30 min before end of the dialysis to measure dp-ucMGP.

The presence of vitamin K_1_ was tested in six samples of dialysate—two samples from three different participants. One sample was collected 20–30 min after dialysis start and one 20–30 min before end of treatment.

### 2.3. Absorption Capacity

After a 12 h fast, seven patients ingested 15 g D-xylose dissolved in 150 mL water and after that another 150 mL of water. D-xylose was measured in the blood after 60 min. The plasma concentration of D-xylose was measured with the ‘D-xylose Assay Kit’ from Megazyme^©^ on Pentra 400 (HORIBA ABX SAS).

### 2.4. Dp-ucMGP and Protein Intake

Sixteen patients drank Renilon 7.5 Nutricia^®^, 125 mL a day (contains 9 g protein, 249 kcal and 14 µg vitamin K) for 14 days as a protein supplementation specially developed for patients with kidney failure. dp-ucMGP was measured in plasma at the beginning of this study and two weeks later. At the same time, the patients answered a FFQ focused on protein intake. The protein FFQ was based on the data form Frida, a food database by Technical University of Denmark (DTU) [20] and USDA Food Composition Databases [18].

### 2.5. Biochemistry

dp-ucMGP samples were analysed in EDTA plasma at Odense University Hospital using IDS-iSYS inaKtif MGP analysis. The range of measurement was 300–12,000 pmol/L and the reference measure was 750 pmol/L (IDS, 2017b) [21]. The reference was based on a study of 132 healthy individuals. Their dp-ucMGP values ranged from <300 to 824 pmol/L, with 95% of the values below 750 pmol/L. This method was used for both plasma and dialysate samples.

Vitamin K_1_ was analysed at DTU by use of a LC–MS/MS method, as previously described [22].

### 2.6. Statistics

Data was presented as the mean (±standard deviation), median (range) and number (percentages) as appropriate. The Wilcoxon Rank-Sum Test was used to test for paired differences in substudy 2–4 and the Mann–Whitney Rank-Sum Test for unpaired tests for confounding in substudy 1. Spearman’s correlation was used to explore the relation between intake of vitamin K and levels of plasma dp-ucMGP.

A *p*-value below 0.05 was considered as statistically significant. All statistical calculations were performed using SPSS (v.25, IBM, SPSS Statistics).

The Regional Committee have ethics approval for the investigation (no. H-17036789) and the Danish Authorities for data protection. The investigation fulfilled the conditions in the Agreement of Helsinki.

## 3. Results

Baseline characteristics of the participants are presented in Table 1.

### 3.1. Vitamin K Status and Intake

The intake of phylloquinone, menaquinone and total vitamin K was calculated based on the reported FFQ. Results are shown in Table 2. Phylloquinone was the main source of vitamin K with a median intake of 54 μg/d.

dp-ucMGP in plasma was used as an indicator of vitamin K status and 750 pmol/L was used as reference value for vitamin K deficiency [23]. Plasma dp-ucMGP concentrations before dialysis treatment were 1856 pmol/L (median) and every participant was above the reference value of 750 pmol/L. These results imply that all the patients had vitamin K deficiency. Concentrations varied from 914 to 5635 pmol/L (Figure 1). We found no correlation between the intake of vitamin K as assessed by a FFQ and plasma dp-ucMGP (*r* = 0.120; *p* = 0.536).

### 3.2. Washout of Vitamin K (Phylloquinone) and dp-ucMGP

dp-ucMGP concentrations were measured in plasma before and after haemodialysis treatment. dp-ucMGP concentrations were significantly lower after dialysis treatment (Table 3). Moreover, dp-ucMGP concentrations were measured in dialysate. These results show that dp-ucMGP were present in dialysate in approximately the same concentrations at the beginning and the end of haemodialysis treatment (Table 3). Phylloquinone was not present in the dialysate (Table 3).

### 3.3. Absorption Capacity

D-xylose concentration in serum (s-D-xylose) one hour after consumption of D-xylose ranged from 1285 to 4534 μmol/L, with 5 of 7 patients above the reference value of 1769 μmol/L [24] (Figure 2). The remaining two patients had values close to the reference value. The median value after one hour was 2245 μmol/L. D-xylose concentrations in serum before consumption of D-xylose were close to zero (0–108 μmol/L).

### 3.4. dp-ucMGP and Protein Intake

Plasma dp-ucMGP concentrations before and after consumption of one bottle of Renilon 7.5^®^ daily for 14 days were compared to measure whether plasma dp-ucMGP concentrations were affected by the increased intake of protein in the short term. The concentrations were not significantly different (Table 4).

Protein intake at baseline was calculated from a FFQ as g/kg body weight/day. Median intake was below the recommended 1 g/kg/d for men and women in haemodialysis. The results are presented in Table 5.

## 4. Discussions

The present study confirms that haemodialysis patients have an increased concentration of plasma dp-ucMGP, suggesting a low vitamin K status. This is consistent with the previously published results [11,12].

Our results do not suggest that washout of phylloquinone during haemodialysis is of clinical relevance as an explanation for low vitamin K status. Phylloquinone was not present in the dialysate, indicating that phylloquinone is not washed out during dialysis treatment. This result was expected, as phylloquinone is a lipophile vitamin. Washout of menaquinones was not tested but as this form of vitamin K is even more lipophilic [25], we do not expect this to be washed out either.

A low absorption of vitamin K could potentially explain a low vitamin K status. The theory was that fluid accumulation between haemodialysis treatments might cause intestinal oedema and affect absorption. However, we did not find any results indicating a decreased absorption capacity.

The average intake of vitamin K in the Nordic population is estimated to be 120 μg/d [26]. The median intake of total vitamin K in our participants was 90 μg/d, with a range of 7–259 μg/d. However, we found no correlation between vitamin K intake estimated by a FFQ and dp-ucMGP concentrations. Not many studies report data on vitamin K1 and vitamin K2 estimated by a FFQ [26]. In a Norwegian study, the estimated dietary intake was 130 µg phylloquinone/d and 15–20 µg menaquinones/d. Thus, our data seems lower for phylloquinone but almost similar for menaquinone. This is in accordance with the dietary restrictions of vegetable and fruit intake high in phylloquinones in dialysis patients. It is challenging to estimate the dietary intake of the longer menaquinone, as this are not available in food databanks, and the content of vitamin K in foods may differ between regions and countries.

To our knowledge, washout of dp-ucMGP and vitamin K during haemodialysis treatment has never been measured before and neither has protein intake in relation to plasma dp-ucMGP. Therefore, we explored these issues.

A reduced intake of protein may affect the ability to synthesise MGP in the body. Increased protein intake in terms of an additional 9 g of protein a day for 14 days did not change the levels of plasma dp-ucMGP in the present study. However, the overall amount of consumed protein was far below the recommended intake for haemodialysis patients, and higher amounts of protein supplementation for a longer period may have affected the results. A larger sample size may also have shown some changes not detectable in this smaller group of patients. In addition, we did not examine whether the active form of MGP was affected by a change in the protein intake.

The size of the dp-ucMGP molecule is reported to be between 10 and 14 kDa. The manufacturer of the dialysis membrane used for haemodialysis treatment at our dialysis unit (Revaclear 400, Baxter) states that their membrane removes molecules up to 11.8 kDa from the blood [27]. Therefore, it was possible that dp-ucMGP might be detectable in the dialysate. Moreover, the molecule may adhere to the membrane and tubes and thus be removed during haemodialysis. The dp-ucMGP concentrations were significantly decreased in plasma after a 3–4 h haemodialysis session. dp-ucMGP was present in the dialysate in approximately the same concentration at the beginning (327 pmol/L) and end of dialysis treatment (414 pmol/L). This suggests a constant washout of dp-ucMGP during haemodialysis. However, the IDS-iSYS inaKtif MGP analysis is designed for plasma samples and not dialysate. This may influence the concentration of dialysate dp-ucMGP, but we estimate that this does not influence the presence of dp-ucMGP. Due to the high levels of pre-dialysis plasma dp-ucMGP, the plasma concentrations must be restored between each dialysis session, either by new synthesis of dp-ucMGP or exchange from the intracellular phase to plasma between the dialysis sessions.

As dp-ucMGP is affected by the dialysis treatment, the phosphorylated, carboxylated active form of MGP may also be lost during dialysis. If the active form of MGP is lost during dialysis treatment as well, this could theoretically accelerate the development of vascular calcification.

## 5. Conclusions

We confirm that haemodialysis patients have a very low vitamin K status but found no indications that poor absorption capacity or washout during haemodialysis was responsible for this. A low vitamin K intake is the most likely reason for the deficiency of vitamin K.

Plasma dp-ucMGP (the biomarker of vitamin K status) was washed out during haemodialysis, but the clinical implications of this are unclear. Protein supplementation did not affect the levels of plasma dp-ucMGP

Further studies on how to improve vitamin K intake in dialysis patients are warranted.

## Figures and Tables

**Figure 1 nutrients-12-02513-f001:**
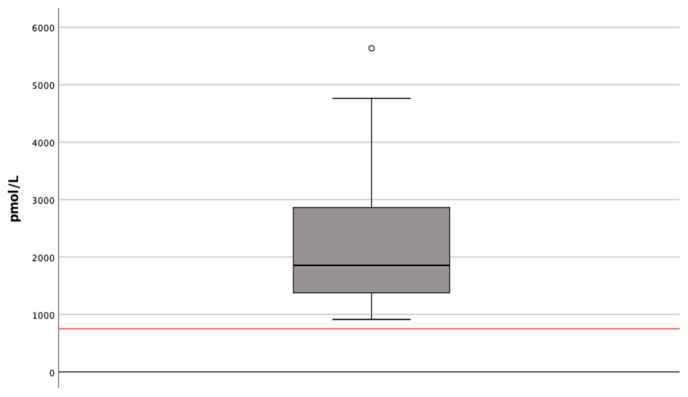
Plasma dp-ucMGP concentrations before dialysis treatment. The reference value (750 pmol/L) is marked with a red horizontal line (*N* = 30). The boxplot shows the median, the interquartile range and the ranges. ° represents an outlier

**Figure 2 nutrients-12-02513-f002:**
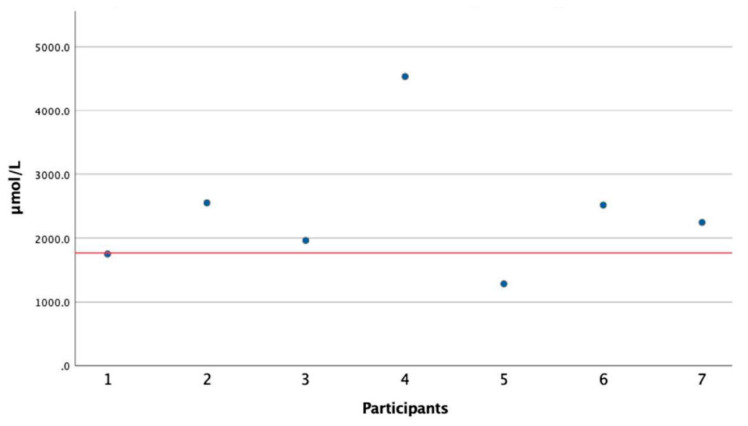
D-xylose concentrations in serum one hour after consumption of 15 g of D-xylose. The reference value for normal intestinal absorption is marked with a red horizontal line.

**Table 1 nutrients-12-02513-t001:** Baseline characteristics of the participants in the four substudies on vitamin K deficiency.

Substudy	1	2	3	4
Number of participants	30	16	7	16
Age (mean ± standard deviation)	66 ± 10	68 ± 12	70 ± 10	66 ± 9
Females (%)	27	25	14	19
BMI (mean ± standard deviation)	24.8 ± 10.9	26.7 ± 6.1	25.3 ± 3.7	25.0 ± 12.2
Dialysis vintage [month] median (range)	28 (6–290)	24 (6–150)	36 (15–103)	19 (6–71)
Diabetes (%)	40	69	0	50
Causes of end-stage kidney disease (%)				
Diabetic nephropathy	17	44	0	25
Multiple myeloma	10	13	0	6
Polycystic kidney disease	17	6	29	6
Glomerulonephritis	17	0	29	6
Nephropathy not otherwise specified	22	25	42	38
Other	17	12	0	19

BMI: Body Mass Index.

**Table 2 nutrients-12-02513-t002:** Vitamin K intake in patients on haemodialysis based on a Food Frequency Questionnaire (FFQ).

	Phylloquinone Median (Range)	Menaquinone Median (Range)	Total Vitamin K Median (Range)
Daily intake [μg/d]	54 (1–234)	26 (6–134)	90 (7–259)

**Table 3 nutrients-12-02513-t003:** Vitamin K-related variables in dialysate and plasma.

	Start	End	Δ dp-ucMGP	*p*-Value
Dialysate dp-ucMGP [pmol/L] (median (range))	327 (305–630)	414 (323–520)	5 (−247–154)	0.569
Dialysate phylloquinone [ng/mL]	<0.01 *	<0.01 *	0	-
Plasma dp-ucMGP [pmol/L] (median (range))	1856 (914–5635)	849 (485–1785)	−1022 (−3850–−326)	<0.0001

Samples were collected at the beginning (20–30 min after treatment start) and end (20–30 min before end of treatment) of a haemodialysis session. dp-ucMGP: dephosphorylated-uncarboxylated Matrix Gla Protein. * Detection limits given instead of 0.

**Table 4 nutrients-12-02513-t004:** Plasma dp-ucMGP concentrations before and after 14 days of Renilon supplement in haemodialysis patients—differences in concentrations are shown.

	Plasma dp-ucMGP before [pmol/L]	Plasma dp-ucMGP after [pmol/L]	Δ Plasma dp-ucMGP	*p*-Value
**Men** (*n* = 13) (median (range))	2015 (914–4764)	1833 (449–4016)	−23 (−1690–373)	0.35
**Women** (*n* = 3) (median (range))	3338 (1922–5635)	1891 (1185–4944)	−737 (−1447–−691)	0.11
**Total** (*n* = 16) (median (range))	2077 (914–5635)	1851 (449–4944)	−165 (−1690–373)	0.06

**Table 5 nutrients-12-02513-t005:** Protein intake calculated from a FFQ in 16 patients treated with haemodialysis.

	Men (*n* = 13)	Women (*n* = 3)	Total (*n* = 16)
Protein intake [g/kg/d] (median (range))	0.5 (0.05–0.84)	0.72 (0.4–0.84)	0.6 (0.05–0.84)

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
