# Peer review of "Causes of Vitamin K Deficiency in Patients on Haemodialysis"

_nutrients, 2020, doi:10.3390/nu12092513_

Round 1

Reviewer 1 Report

The paper by Wikstrom et al describes the results of a study aiming at investigating possible underlying causes of low vitamin K status in patients on haemodialysis. The authors investigated if this could be due to a low vitamin K intake, washout during dialysis or inhibited absorption capacity. Overall the study is correctly presented and data is credible.The results suggest that the low vitamin K status is not due to removal of vitamin K during dialysis or decreased absorption, but is likely due to low intake of vitamin K in the food.

This is an interesting study that, although not using a large number of subjects, provides information of interest to the field and contributes to support the statement that low vitamin K intake, rather then its loss, may be the major cause of accelerated atherosclerosis in patients undergoing haemodialysis.

Some minor english spell checks are required.

Author Response

The manuscript has been spell checked by a native English speaking nephrologist

Reviewer 2 Report

interesting study by the authors, and potentially an important additional deficient dietary factor whose replacement could be of help to many HD pts.

i only have a few comments:

1) table 1: says that 10% of the pts were on HD due to myelomatosis?  do you mean 3-4 pts of this cohort had a plasma cell malignancy?  that seems very unusual.  did no one in the cohort have ESRD from high BP?

2) substudy 4, table 4.  for those of us who dont know what Renilon is, it would be helpful to explain that it contains 10 mcg of some kind of vitamin K.

3) exclusion criteria:  beside rx with warfarin, were there other exclusion criteria for your study?

4) sample size:  given the small number of people in your studies, do you think the sample size could have impacted your results?  this should be mentioned in your discussion section. 

5) future studies:  based on your research, what would be your next steps?

Author Response

Thank you for your nice comments to our exploratory studies.

1)

We agree that 10 % is a high fraction of patients with myelomatosis. However, this is the correct number. This may be due to the large department of haematology at our hospital taking care of patients with myelomatosis.

2)

This is of great importance. Thank you. The vitamin K content has been added to method section page 3 line 111-112

3)

Known malabsorptive diseases and recent intake of vitamin K supplements were also exclusion criteries. Patients with diabetes were not included in study 3. Page 2 line 77-79

4)

We certainly agree. The limitation due to the small sample size is mentioned at page 7 line 262-263

5)

As this indicates a low intake of vitamin K to cause the vitamin K deficiency, we would like to investigate ways to improve the vitamin K intake in this population. This perspective has been added at page 7 line 288